# *Limosilactobacillus fermentum* Strain 3872: Antibacterial and Immunoregulatory Properties and Synergy with Prebiotics against Socially Significant Antibiotic-Resistant Infections of Animals and Humans

**DOI:** 10.3390/antibiotics11101437

**Published:** 2022-10-19

**Authors:** Vyacheslav M. Abramov, Igor V. Kosarev, Andrey V. Machulin, Tatiana V. Priputnevich, Irina O. Chikileva, Evgenia I. Deryusheva, Tatiana N. Abashina, Almira D. Donetskova, Alexander N. Panin, Vyacheslav G. Melnikov, Nataliya E. Suzina, Ilia N. Nikonov, Marina V. Selina, Valentin S. Khlebnikov, Vadim K. Sakulin, Raisa N. Vasilenko, Vladimir A. Samoilenko, Vladimir N. Uversky, Andrey V. Karlyshev

**Affiliations:** 1Federal Service for Veterinary and Phytosanitary Surveillance (Rosselkhoznadzor), Federal State Budgetary Institution “The Russian State Center for Animal Feed and Drug Standardization and Quality” (FGBU VGNKI), 123022 Moscow, Russia; 2Kulakov National Medical Research Center for Obstetrics, Gynecology and Perinatology, Ministry of Health, 117997 Moscow, Russia; 3Skryabin Institute of Biochemistry and Physiology of Microorganisms, Federal Research Center “Pushchino Scientific Center for Biological Research of Russian Academy of Science”, Russian Academy of Science, 142290 Pushchino, Russia; 4Laboratory of Cell Immunity, Blokhin National Research Center of Oncology, Ministry of Health RF, 115478 Moscow, Russia; 5Institute for Biological Instrumentation, Federal Research Center “Pushchino Scientific Center for Biological Research of Russian Academy of Science”, Russian Academy of Science, 142290 Pushchino, Russia; 6NRC Institute of Immunology FMBA of Russia, 115522 Moscow, Russia; 7Gabrichevsky Research Institute for Epidemiology and Microbiology, 125212 Moscow, Russia; 8Federal State Educational Institution of Higher Professional Education, Moscow State Academy of Veterinary Medicine and Biotechnology named after K.I. Skryabin, 109472 Moscow, Russia; 9Institute of Immunological Engineering, 142380 Lyubuchany, Russia; 10Department of Molecular Medicine, Morsani College of Medicine, University of South Florida, Tampa, FL 33612, USA; 11Department of Biomolecular Sciences, Faculty of Health, Science, Social Care and Education, Kingston University London, Kingston upon Thames KT1 2EE, UK

**Keywords:** *L. fermentum*, antibiotic resistance, prebiotics, probiotics, antibacterial activity, immunoregulation, *S. aureus*, *E. coli*, *Salmonella*, *Campylobacter*

## Abstract

*Limosilactobacillus fermentum* strain 3872 (LF3872) was originally isolated from the breast milk of a healthy woman during lactation and the breastfeeding of a child. The high-quality genome sequencing of LF3872 was performed, and a gene encoding a unique bacteriocin was discovered. It was established that the bacteriocin produced by LF3872 (BLF3872) belongs to the family of cell-wall-degrading proteins that cause cell lysis. The antibacterial properties of LF3872 were studied using test cultures of antibiotic-resistant Gram-positive and Gram-negative pathogens. Gram-positive pathogens (*Staphylococcus aureus* strain 8325-4 and *S. aureus* strain IIE CI-SA 1246) were highly sensitive to the bacteriolytic action of LF3872. Gram-negative pathogens (*Escherichia coli*, *Salmonella* strains, and *Campylobacter jejuni* strains) were more resistant to the bacteriolytic action of LF3872 compared to Gram-positive pathogens. LF3872 is a strong co-aggregator of Gram-negative pathogens. The cell-free culture supernatant of LF3872 (CSLF3872) induced cell damage in the Gram-positive and Gram-negative test cultures and ATP leakage. In the in vitro experiments, it was found that LF3872 and Actigen prebiotic (Alltech Inc., Nicholasville, KY, USA) exhibited synergistic anti-adhesive activity against Gram-negative pathogens. LF3872 has immunoregulatory properties: it inhibited the lipopolysaccharide-induced production of proinflammatory cytokines IL-8, IL-1β, and TNF-α in a monolayer of Caco-2 cells; inhibited the production of IL-12 and stimulated the production of IL-10 in immature human dendritic cells; and stimulated the production of TGF-β, IFN-γ, and IgA in the immunocompetent cells of intestinal Peyer’s patches (PPs) in mice. These results indicate the possibility of creating a synbiotic based on LF3872 and a prebiotic derived from *Saccharomyces cerevisiae* cell wall components. Such innovative drugs and biologically active additives are necessary for the implementation of a strategy to reduce the spread of antibiotic-resistant strains of socially significant animal and human infections.

## 1. Introduction

The long-term use of antibiotics in industrial animal husbandry as growth factors and inhibitors of intestinal pathogens has led to the emergence of antibiotic-resistant strains and their spread among animals and the world population consuming agricultural products [1,2]. Antibiotics introduced into farm animal feed have increased the prevalence of antibiotic-resistant strains of microorganisms (*Escherichia*, *Salmonella*, and *Campylobacter*) belonging to the phylum of proteobacteria [3,4]. The consumption of agricultural products obtained using feed containing antibiotics has caused an increase in the human intestinal microbiota of strains belonging to the proteobacteria phylum, inducing the development of intestinal dysbiosis and resulting in a decrease in intestinal barrier function, the development of chronic systemic inflammation, damage to the cardiovascular system, and the development of metabolic syndrome [5,6,7]. In women of reproductive age, intestinal dysbiosis stimulates the development of reproductive system dysbiosis and the growth of urogenital infections [8]. As a result, more than one billion women in the world each year turn to clinics for the treatment of urogenital infections [9].

The long-term use of antibiotics has led to the emergence and spread of antibiotic-resistant strains of *E. coli* around the world. A pandemic of extraintestinal infectious diseases caused by these strains of *E. coli* (CTX-M-15, ST131-H30R, ST1193, FQR, TEM, and SHV) persisting in the human intestine was first reported in 2008 and continues to develop [10,11,12,13]. Pathogenic multidrug-resistant (MDR) strains of *E. coli* isolated from broilers belonging to the clone ST 131 have genetic similarity and common virulence genes with *E. coli* isolates circulating in the human intestine and causing extraintestinal infectious diseases (cystitis, pyelonephritis, meningitis, and sepsis) [14]. Cystitis or pyelonephritis in pregnant women is one of the main pathogenetic causes of preterm birth and necrotizing enterocolitis (NEC) in newborns [15]. The resistance of *E. coli* strains to antimicrobial drugs is mainly mediated by extended-spectrum β-lactamases (ESBLs) of the TEM, SHV, and CTX-M-15 classes [16,17]. The spread of specific resistant clonal groups of *E. coli* largely explains the cause of this pandemic [18,19]. The main H30 subclone of *E. coli*, clone 131 (ST131-H30), belonging to the phylogroup B2 and containing genes encoding class CTX-M-15 ESBL and the type 1 fimbrial adhesins FimH or UclD, became the leading line of resistance to two key classes of antimicrobial drugs for the treatment of extraintestinal infectious diseases—fluoroquinolones and extended-spectrum cephalosporins. This led to the rapid development of a pandemic circulating in farm animals and humans [10,19,20,21,22].

*Salmonella* are zooanthroponotic pathogens transmitted from animals to humans, and then from humans to humans [23]. Since the mid-1980s, the dominant *Salmonella enterica* serovar Typhimurium (*S*. Typhimurium) has been overtaken by the highly virulent antibiotic-resistant *Salmonella enterica* serovar Enteritidis (*S*. Enteritidis), which currently dominates salmonella infections in children and adults worldwide [24,25]. An important factor in the virulence of *S.* Enteritidis is the ability to form a biofilm in the intestines of animals and humans under anaerobic conditions [26]. This biofilm increases the antibiotic resistance of the pathogen and stimulates the production of virulence factors [27]. *S.* Enteritidis strains are able to colonize the lymph nodes and preovulatory follicles of poultry, causing a decrease in egg production and an increased risk of foodborne salmonellosis in humans [28,29]. *S.* Enteritidis strains have caused numerous large-scale national and international outbreaks of intestinal diseases with complex transmission routes [30,31,32,33]. Strains of *S.* Enteritidis causing extraintestinal infections carry plasmids that confer MDR and ensure the spread of these strains throughout the world, posing a serious threat to global health and agriculture [34,35].

*Campylobacter* is the most common causative agent of human gastroenteritis in the world, and *C. jejuni* is responsible for more than 80% of cases [7,36,37]. In the EU, campylobacteriosis has been the most frequently reported cause of gastroenteritis in humans since 2005 [38]. The European Food Safety Authority has reported that *C. jejuni* is present in 86% of chicken carcasses across Europe [39]. The consumption of industrial poultry products is the main route of the transmission of this pathogen to humans [40,41]. In humans, the manifestations of *C. jejuni* infection range from asymptomatic carriage to bloody diarrhea, fever, and abdominal pain, as well as serious post-infection sequelae, such as neuromuscular palsy (Guillain–Barré syndrome) [42]. *C. jejuni* infects newborns, causing NEC, which is characterized by high infant mortality rates. Older children with campylobacteriosis experience growth retardation and lifelong physical and cognitive impairments [43]. WHO data indicate the relevance of the development of new preventive and therapeutic agents against *C. jejuni* [44].

Some probiotics reduce the colonization of the intestine by pathogenic *E. coli*, including antibiotic-resistant strains [45]; reduce the colonization of the intestine by *Salmonella* and prevent biofilm formation [46,47]; and reduce the colonization of the intestine by *Campylobacter* [48]. However, there is an emergent need for further research to find more effective probiotics, prebiotics, metabiotics, and synbiotics for the prevention of infectious diseases caused by antibiotic-resistant strains of *E. coli*, *Salmonella*, and *Campylobacter* in farm animals and humans. In order to effectively overcome the resistance of *E. coli*, *C. jejuni*, and *Salmonella* pathogens and develop rational therapies for infectious conditions, it is necessary to search for new antimicrobial agents. One of the modern trends in this field is the investigation of the antimicrobial potential of probiotic lactic acid bacteria producing peptidoglycan-degrading bacteriocins, as well as enzymes that increase the permeability of the outer membrane (OM) of Gram-negative microorganisms. *Limosilactobacillus fermentum* strain 3872 (LF3872) was originally isolated from the milk of a healthy woman during lactation and the breastfeeding of a child [49,50,51,52]. The complete whole genome sequencing of the LF3872 strain revealed a gene encoding a unique bacteriocin, which was absent in all probiotic lactobacilli with the currently established genome structure [53].

The use of prebiotics, such as carbohydrates isolated from *Saccharomyces cerevisiae* cellular walls (mannan-oligosaccharides (MOS), fructo-oligosaccharides (FOS), and β-glucans), has been the focus of several studies in poultry [54,55,56,57,58]. These prebiotics can inhibit the adhesion of *Escherichia*, *Salmonella*, and *Campylobacter* to enterocytes [59].

The aim of this work was to use bioinformatics tools to compare the bacteriocin produced by LF3872 (BLF3872) with the class III bacteriocins, such as enterolysin A from *Enterococcus faecalis* LMG 2333, zoocin A from *Streptococcus equi* subsp. *zooepidemicus*, and zoocin A peptidase family M23 from *Flavobacterium johnsoniae.* Enterolysin A and the different types of zoocins A [60] are proteins belonging to the M37/M23 family of metallopeptidases [61]. Members of this family are endopeptidases that lyse bacterial cell walls [62]. All of these proteins have been grouped as class III bacteriocins and are known as bacteriolysins. Additionally, we studied the antibacterial, co-aggregative, and immunoregulatory properties of LF3872 and the synergism of the anti-adhesive activity of Actigen prebiotic (Alltech Inc., USA) with the cell-free culture supernatant of LF3872 (CSLF3872).

## 2. Results

### 2.1. Comparison of Bacteriocin BLF3872 with Class III Bacteriocins

The profile of the peptidase M23 domain in class III bacteriocins was determined according to the Pfam database. Proteins with this domain are characterized as zinc metallopeptidases and endopeptidases with a range of specificities. The percentages of identity (ClustalO) and the pairwise distances (MEGAX) in the 280–404 region between the BLF3872 bacteriocin sequence and the peptidase M23 domain from enterolysin A, zoocin A, and zoocin A peptidase family M23 are shown in Table 1.

The pairwise distances [63] for the investigated sequences confirmed the positive selection between the peptidase M23 domains in the investigated class III bacteriocins and the region in the BLF3872 bacteriocin. The results of the logo motif search in the 280–404 region in the BLF3872 bacteriocin corresponding to the peptidase M23 domain and peptidase M23 domain from enterolysin A, zoocin A, and zoocin A peptidase family M23 are shown in Figure 1.

For enterolysin A, the N-terminal domain is expected to be responsible for the catalytic activity of an enzyme that will cleave chemical bonds in the peptidoglycan of the bacterial cell wall [65]. In zoocins, the N-terminal domain plays a catalytic role in hydrolyzing the D-alanyl-L-alanine peptide bond in susceptible peptidoglycan, leading to bacterial cell lysis [65], wherein the single Zn^2+^ ion is coordinated by two conserved motifs HXXXD and HXH (with X being any amino acid) [66]. Residues His280 and Asp284 of the BLF3872 bacteriocin were present in the HXXXD motif, and residues His361 and His363 were present in the HXH motif (Figure 1). These results indicated that the BLF3872 bacteriocin showed homology to cell-wall-degrading proteins that cause bacterial cell lysis. Thus, BLF3872 is a member of the class III bacteriocins and belongs to the group of bacteriolysins that destroy the peptidoglycan of the cell wall of bacterial pathogens.

### 2.2. Antibacterial Activity of LF3872 against Staphylococcus aureus

The antagonistic activity of LF3872 against pathogenic *S. aureus* strains is shown in Table 2. In the co-culture experiments, the Gram-positive pathogens *S. aureus* strain 8325-4 and *S. aureus* strain IIE CI-SA 1246 were highly sensitive to the bacteriolytic action of LF3872. The co-cultivation of *S. aureus* strains together with LF3872 for 24 h reduced the level of living test cultures by six log. Heat-treated LF3872 did not reduce the level of live test cultures of *S. aureus*.

### 2.3. Antibacterial Activity of LF3872 against E. coli Pathogens

The antagonistic activity of LF3872 against pathogenic *E. coli* strains is shown in Table 3. The cultivation of *E. coli* strains, including ESBL-containing strains, enteropathogenic (EPEC) strains, and enterotoxigenic (ETEC) strains together with LF3872 for 24 h reduced the level of living test cultures by three log. The cultivation of *E. coli* strains together with LF3872 for 48 h reduced the level of living test cultures by four log. Heat-treated LF3872 did not reduce the level of live test cultures of *E. coli*.

### 2.4. Antibacterial Activity of LF3872 against S. Enteritidis and S. Typhimurium Strains

The antagonistic activity of LF3872 against *Salmonella* strains is shown in Table 4. The cultivation of *S.* Enteritidis strains and *S.* Typhimurium strains together with LF3872 for 24 h reduced the level of living cells in the test cultures by three log. The cultivation of *S.* Enteritidis strains and *S.* Typhimurium strains together with LF3872 for 48 h reduced the level of living test cultures by four log. Heat-treated LF3872 did not reduce the level of live test cultures of *Salmonella.*

### 2.5. Antibacterial Activity of LF3872 against C. jejuni Strains

The antagonistic activity of LF3872 against *C. jejuni* strains is shown in Table 5. The cultivation of *C. jejuni* strains together with LF3872 for 24 h reduced the level of living test cultures by two log. The cultivation of *C. jejuni* strains together with LF3872 for 48 h reduced the level of living cells in the test cultures by three log. Heat-treated LF3872 did not reduce the level of live test cultures of *C. jejuni*.

### 2.6. CSLF3872 Induces ATP Leakage from Cells of Test Cultures

CSLF3872 induced cell damage in Gram-positive and Gram-negative test cultures and ATP leakage. The levels of ATP leakage were investigated based on non-selective pore formation and indexes of cell injury. The results of the investigations are shown in Table 6. The strains of Gram-positive pathogens were very sensitive to the damaging effect of CSLF3872. The cultivation of *S. aureus* strain IIE-8325-4 in the presence of CSLF3872 for 2.5 h increased the level of extracellular ATP from 6.0 ± 1.3 (nm/OD) to 358.0 ± 14.3 (nm/OD) (*p* < 0.001), and *S. aureus* strain IIE CI-SA 1246 (clinical isolate) increased the level of extracellular ATP from 5.4 ± 1.5 (nm/OD) to 360.5 ± 12.6 (nm/OD) (*p* < 0.001). The strains of Gram-negative pathogens (EPEC *E. coli* ATCC E 2348/69, ETEC *E. coli* ATCC E 31705, *E. coli* ATCC BAA198, *E. coli* ATCC BAA 204, *E. coli* ATCC BAA 2326, *E. coli* IIE Br 5164, *E. coli* IIE Br 5372, *E. coli* IIE Pi 5548, *E. coli* IIE Co 5622, and *E. coli* IIE Hu 4326, causing extra-intestinal infectious diseases in animals and humans; *S.* Enteritidis ATCC 13076; *S.* Enteritidis ATCC 4931; *S.* Enteritidis IIE Egg 6215; *S.* Typhimurium ATCC 700720; *S.* Typhimurium ATCC 14028; and *S.* Typhimurium IIE Br 6458) were more resistant to the damaging effects of CSLF3872. The cultivation of *E. coli* strains and *Salmonella* strains in the presence of CSLF3872 for 2.5 h increased the level of extracellular ATP on average from 4.0 (nm/OD) to 43.1 (nm/OD) (*p* < 0.01). The strains of the Gram-negative pathogen *C. jejuni* (*C. jejuni* ATCC 29428, *C. jejuni* IIE Br 7154, *C. jejuni* IIE Br 7365, and *C. jejuni* IIE Br 7548) were the most resistant to the damaging effects of CSLF3872. The cultivation of *C. jejuni* strains in the presence of CSLF3872 for 2.5 h increased the level of extracellular ATP on average from 5.8 (nm/OD) to 20.4 (nm/OD) (*p* < 0.05).

### 2.7. Cells of LF3872 Co-Aggregate with Proteobacteria Pathogens

We evaluated the co-aggregation capacities of LF3872 with strains of proteobacteria pathogens (*E. coli*, *Salmonella*, and *Campylobacter*). At pH 7.0 in phosphate-buffered saline (PBS) buffer, LF3872 co-aggregated with all strains of *E. coli*, *Salmonella*, and *Campylobacter*. The percentage of co-aggregation ranged from 32.5 ± 3.4% to 43.3 ± 4.5%. At pH 5.0 in 2-(N-morpholino)ethanesulfonic acid (MES) buffer, LF3872 also co-aggregated with all studied strains of *E. coli*, *Salmonella*, and *Campylobacter* pathogens. At a low pH value, the percentage of co-aggregation significantly increased (*p* < 0.05), ranging from 64.2 ± 5.6% to 75.8 ± 6.7% (Table 7). LF3872 is a strong co-aggregator of *E. coli*, *Salmonella*, and *Campylobacter* pathogens. Thus, the co-aggregation ability of LF3872 may contribute to preventing the colonization of the human intestinal mucosal surface by proteobacteria pathogens.

### 2.8. Synergism of CSLF3872 and Actigen Prebiotic in Inhibiting the Adhesion of Proteobacteria Pathogens (E. coli, Salmonella, Campylobacter) to Caco-2 Cells

To study whether CSLF3872 has an effect on the adhesion of pathogens of proteobacteria (*E. coli*, *Salmonella*, and *Campylobacter*), the cell-free supernatant was incubated jointly with pathogens of proteobacteria, before they were incubated with a monolayer of Caco-2 cells (Table 8). The co-incubation of proteobacteria pathogens with CSLF3872 led to a significant decrease in the adhesion of proteobacteria pathogens to Caco-2 cells (*p* < 0.01). The co-incubation of Actigen prebiotic with proteobacteria pathogens also led to a significant decrease in the adhesion of proteobacteria pathogens to Caco-2 cells (*p* < 0.01) (Table 8). The mixture of Actigen prebiotic and CSLF3872 showed synergism in inhibiting the adhesion of proteobacteria pathogens to Caco-2 cells (*p* < 0.001) (Table 8).

### 2.9. LF3872 Regulates Cytokine Production in Intestinal Caco-2 Cells

LF3872 did not stimulate the production of pro-inflammatory cytokines IL-8, TNF-α, and IL-1β in Caco-2 enterocytes, unlike lipopolysaccharide (LPS) (a TLR4 receptor agonist) and MALP-2 (a TLR-2/6 receptor agonist). When LPS + LF3872 and MALP-2 + LF3872 were co-introduced into the culture medium, there was a decrease in the IL-8 production induced by LPS and MALP-2 (Figure 2A), a decrease in the TNF-α production (Figure 2B), and a decrease in the IL-1β production (Figure 2C). LF3872 slightly increased the IL-6 production (Figure 2D).

### 2.10. LF3872 Regulates Cytokine Production in Immature Dendritic Cells

Immature dendritic cells (IDCs) function as biosensors of innate immunity and provide immune homeostasis. IDCs were obtained from human peripheral blood monocytes. Human IDCs were subjected to light microscopy analysis after azure-eosin staining (Figure 3A). IDC phenotyping was performed by flow cytometry using labeled monoclonal antibodies (Beckman Coulter, USA). The IDCs, unlike the primary blood monocytes, did not express CD83 (Figure 3B) and CD14 on their surface (Appendix A). The IDCs had a high expression of HLA-DR antigens (91 ± 6%) and a high expression of CD-80 costimulatory molecules (65 ± 5%) (Figure 3B). LF3872 stimulated the production of IL-10 in IDCs, which prevented the paracrine production of proinflammatory cytokines. In addition, IDCs in the presence of LPS (TLR4 agonist receptors of innate immunity) produced IL-12, which activated natural killer (NK) cells and natural killer T (NKT) cells and ensured the development of a pro-inflammatory immune response, disrupting immune homeostasis. LF3872 inhibited the production of IL-12 in IDCs to the initial control level, preventing the development of an inflammatory reaction. In our experimental conditions, LPS increased the level of IL-12 in the IDC culture medium to 108 ± 2 pg/mL. In the presence of LF3872 in the culture medium, the LPS-induced concentration of IL-12 did not exceed the control level of 25 ± 1 pg/mL. Thus, to ensure immune homeostasis, LF3872 regulated the level of cytokines in the IDCs, stimulating the production of IL-10 and inhibiting the production of IL-12 (Figure 3).

### 2.11. LF3872 Stimulates the Production of TGF-β, IFN-γ Cytokines, and IgA Antibodies in Peyer’s Patches and IgA Antibodies in the Intestines of Mice

LF3872 regulates adaptive immunity. When administered orally to mice, LF3872 stimulated the production of TGF-β, IFN-γ cytokines, and IgA antibodies by immunocompetent Peyer’s patches (PPs) cells (Figure 4) and IgA antibodies in the intestines (Figure 5).

## 3. Discussion

Strain LF3872 produced the bacteriocin BLF3872, which belongs to the class III bacteriocins. This class includes large heat-labile proteins with a high molecular weight (>30 kDa). Class III bacteriocins are subdivided into two distinct groups: the non-lytic antimicrobial proteins and the bacteriolytic enzymes (or bacteriolysins) [67]. The class III bacteriocins helveticin J from *L. helveticus* [68] and helveticin M from *L. crispatus* [69] belong to the non-lytic antimicrobial protein group. Lysostaphin from *S. simulans* biovar *staphylolyticus* [70], enterolysin A from *E. faecalis* LMG 2333 [61], and zoocin A from *S. zooepidemicus* [60] and *F. johnsoniae* are bacteriolysins [61]. Members of this group are endopeptidases that lyse the peptidoglycan present in bacterial cell walls [62]. The obtained results indicated that the BLF3872 bacteriocin showed homology with cell-wall-degrading proteins that cause bacterial cell lysis. Thus, BLF3872 is a member of the class III bacteriocins and belongs to the group of bacteriolysins that destroy the peptidoglycan in the cell walls of bacterial pathogens. Since we isolated strain LF3872 from human milk, at the beginning of our antibacterial experiments we studied its antibacterial properties in a co-cultivation test with two mastitis-inducing antibiotic-resistant strains of *S. aureus*. Staphylococci are the main etiological agents of infectious mastitis lactation [71]. The co-cultivation of LF3872 with two *S. aureus* strains for 24 h reduced the number of living cells of each test culture by six log. In Gram-positive bacteria, the cell wall is a homogeneous electron-dense layer 20–80 nm thick. The bulk of this cell wall (50–90% of the dry matter) is made up of peptidoglycan, which forms a thick, rigid layer. The peptidoglycan layer is permeated with teichoic acids, which can reach the surface of the cell wall. In addition to these basic polymers, the cell walls of Gram-positive bacteria contain small amounts of lipids, polysaccharides, and proteins. Lipids and polysaccharides covalently bind to peptidoglycan, forming a complex, mechanically strong structure [72]. LF3872 produces the peptidoglycan-degrading bacteriocin BLF3872, which can freely enter the cell wall of *S. aureus*, reach the peptidoglycan layer, destroy it, and cause the death of pathogen cells.

In co-cultivation experiments involving LF3872 and different Gram-negative strains of *E. coli*, *S.* Enteritidis, *S.* Typhimurius, and *C. jejuni*, the latter appeared to be more resistant. When co-culturing LF3872 with test cultures of *E. coli*, *S.* Enteritidis, and *S.* Typhimurium for 48 h, the level of living cells of each strain in the test cultures decreased by four log. When co-culturing LF3872 with *C. jejuni* test cultures for 48 h, the level of living cells of each strain in the test cultures decreased by three log. Unlike Gram-positive microorganisms, Gram-negative microorganisms contain an OM in the cell wall. The OM of Gram-negative bacteria provides the cell with an effective permeability barrier against external agents. In most Gram-negative bacteria, the OM is an asymmetric bilayer of phospholipid and LPS, the latter exclusively found in the outer leaflet [73]. CSLF3872 induces cell damage in Gram-positive and Gram-negative pathogens and ATP leakage.

LF3872 is a strong co-aggregator of *E. coli*, *Salmonella,* and *Campylobacter* pathogens. Co-aggregation abilities are characteristic attributes of some probiotic strains belonging to the *Lactobacillaceae* family. Due to these properties, the host organism can avoid the colonization of the intestinal tract by EPEC bacteria [74,75,76,77]. The co-aggregation ability of LF3872 did not depend on the antibiotic resistance of the pathogen strains studied. The co-aggregation increased with a decrease in the pH as a result of an increase in electrostatic interactions.

Actigen prebiotic (Alltech Inc., USA) effectively inhibited the adhesion of *E. coli*, *Salmonella*, and *Campylobacter* to a monolayer of Caco2 cells. Actigen is a mannan-enriched polymer fraction of carbohydrates isolated from the outer layer of the cell wall of the yeast *S. cerevisiae*. The most important functional feature of mannan-oligosaccharides is their ability to interact with receptors on the surface of Gram-negative pathogens (*E. coli*, *Salmonella*, and *Campylobacter*) and block their adhesion to enterocytes and invasion [78]. Actigen prebiotic showed synergy with CSLF3872 in inhibiting the adhesion of these pathogens to a monolayer of Caco-2 cells. Mannose monosaccharide is an established and widely studied ligand for the FimH domain of type I fimbriae. The FimH domain of type I fimbriae is responsible for the recognition of mannose patterns on the host cell exterior and subsequent mannose-dependent pathogenic adhesion [79]. Type I fimbriae are commonly found in *Salmonella* spp. and *E. coli* and play an important role in the adhesion by binding to the mannose patterns in the host cell epithelial receptors [80,81,82]. The glycosides of mannose exhibit amplified anti-adhesive properties towards *E. coli* compared to mannose monosaccharides, indicating the importance of a hydrophobic region in the vicinity of the mannose binding area for type I fimbria adhesion [83,84]. Mannose can bind to the FimH variants from different *E. coli* pathotypes, indicating that the affinity of mannose for the FimH domain is independent of the pathotype [85]. The mannose binding pocket of FimH was later determined to be identical to that found in other pathogenic species, including *E. coli*, *Salmonella*, and *Campylobacter* [86]. The FimH structure varies between species, where this protein is differently presented [86]. These differences are also reflected in the superior specificity of *E. coli* for mono- or trimannose moieties [87,88]. A significant reduction in the adherence of *C. jejuni* to human epithelial cells was observed in the presence of MOS [88]. MOS binds to the FimH domain in competition with the mannose patterns on host epithelial cells. This inhibits pathogenic adhesion by exerting a receptor-mimicking function [89,90]. The inhibition of pathogenic adhesion by MOS is not superior to inhibition by yeast cell walls containing mannose biopolymers [91].

LF3872 inhibited the production of proinflammatory cytokines IL-8, TNF-α, and Il-1β by Caco-2 cells. It is known that inflammation is a rapid development of the protective function of innate immunity [92]. However, the excessive manifestation of the inflammatory response of innate immunity underlies the pathogenetic mechanisms of many maladies, including human and animal diseases [93,94,95,96]. LF3872 increased the production of cytokine IL-6, which is involved in the transformation of the B-lymphocytes into plasma cells that produce immunoglobulin A (IgA) antibodies to neutralize bacteria, toxins, and viruses [97].

Mature dendritic cells (MDCs) are professional antigen-presenting cells that play a central role in ensuring immune homeostasis, including antitumor immunity [98,99]. Due to the high expression of HLA-DR antigens and costimulatory molecules (CD80), MDCs have the ability to activate “naive” T-lymphocytes and induce an antigen-specific immune response. LF3872 stimulated IL-10 production and inhibited IL-12 production in human IDCs, which prevented the spontaneous maturation of these cells and the development of inflammation. MDCs of the mucous membranes of the body form an immune response and regulate the development of inflammation. IDCs function as biosensors of innate immunity and provide immune homeostasis [100].

LF3872, when administered orally to mice, stimulated the production of TGF-β, IFN-γ, and IgA antibodies by the immunocompetent cells of mouse PPs. LF3872 stimulated the production of IgA antibodies in the small intestine and colon. The PPs of the small intestine are the main component of the lymphoid tissue associated with the gastrointestinal tract (gut-associated lymphoid tissue (GALT)), and they act as immune biosensors of the digestive system [101]. In the presence of IL-6 and TGF-β, GALT-associated B-lymphocytes, with the help of the CD4+ T-lymphocytes, turn into plasma cells producing secretory-IgA-isotype antibodies to neutralize bacteria, toxins, and viruses. These plasma cells are disseminated into the lymphoid tissue of various organs, thus making a significant contribution to the protection of the body from infectious agents and toxic substances. Strains of *S. aureus*, *Salmonella*, and *C. jejuni* have the ability to internalize and intercellularly persist in various epithelial cells, including breast epithelial cells and enterocytes [102]. IFN-γ is necessary to ensure the resistance of mucosal epithelial cells to intracellular pathogens. In addition, IFN-γ can exhibit an anti-inflammatory effect, which is important for the prevention and comprehensive treatment of intestinal inflammatory diseases [103].

Despite the fact that enterococci are included in lactic acid bacteria [104], the relationship to these microorganisms is not unambiguous. *E. faecalis* is a resident member of the human intestinal core microbiota harboring a number of pathogenic traits, which explains the association of this bacterium with inflammatory diseases and fatal nosocomial infections [105,106,107]. The microorganisms *S. zooepidemicus* and *F. johnsoniae*, which produce zoocin A peptidase, like *E. faecalis* are also common in opportunistic infections [108]. At the same time, the homologue of enterolysin A, the high-molecular-weight thermolabile bacteriocin BLF3872 produced by LF3872, which has generally recognized as safe (GRAS) status and circulates in the milk of healthy women (in whose genome there are no determinants of pathogenicity, virulence, and antibiotic resistance), will be of great importance for the prevention and comprehensive treatment of lactation mastitis, as well as diseases induced by *S. aureus* and pathogens of the phylum Proteobacteria (*E. coli*, *Salmonella*, and *Campylobacter*).

Our further research will be devoted to the design of a recombinant producer and the development of technology for the isolation of peptidoglycan-degrading bacteriocin BLF3872, as well as further study of its biological properties and mechanisms of action. The design of a recombinant producer and the development of technology for the isolation of peptidoglycan-degrading bacteriocin BLF3872 will further determine the role of the extracellular molecules produced by LF3872 in the culture supernatant and the mechanisms behind their ability to increase the permeability of the OM of Gram-negative pathogens. The mechanisms of the synergistic action of CSLF3872 and prebiotics in relation to the inhibition of pathogen adhesion (*Escherichia*, *Salmonella*, and *Campylobacter*) to enterocytes and uroepitheliocytes will also be studied.

## 4. Materials and Methods

### 4.1. Bacterial Strains and Growth Conditions

LF3872 was originally isolated from the milk of a healthy woman during lactation and the breastfeeding of a child [49,50,51,52]. High-quality whole-genome sequencing of LF3872 was performed [53]. The strain was deposited in the Collection of Microorganisms of the Institute of Immunological Engineering, Lyubuchany, Moscow Region, Russia under the registration number IIE3872. A complete list of microorganisms used in this study and their growth conditions is provided in Table 9.

### 4.2. Identification of BLF3872 Bacteriocin Produced by LF3872

Previously, in [53], the analysis of the complete chromosomal sequences of the LF3872 genome revealed a unique region containing genes encoding a hypothetical protein (835,633–836,847 bp) [53]. This gene is not mentioned in the GenBank nucleotide sequence database but was detected by BAGEL4 (http://bagel.molgenrug.nl/, accessed on 27 September 2022) [111] as a region (830,634–840,633 bp) responsible for the biosynthesis of class III bacteriocins (48% match with Enterolysin A).

### 4.3. Comparative Analysis of BLF3872 Bacteriocin with the Class III Bacteriocins

Sequences of class III bacteriocins were taken from the UniProt database (https://www.uniprot.org/, accessed on 16 October 2022): enterolysin A from *E. faecalis* LMG 2333 (UniProt ID Q9F8B0) [61]; zoocin A from *S. zooepidemicus* (UniProt ID O54308); and zoocin A peptidase family M23 from *F. johnsoniae* (UniProt ID A5FNQ4). The profile of the peptidase M23 domain (PF01551) in the sequences was identified according to the Pfam database (http://pfam.xfam.org/, accessed on 16 October 2022). The multiple sequence alignment of class III bacteriocins and BLF3872 was conducted using the Clustal Omega service (https://www.ebi.ac.uk/Tools/msa/clustalo/, accessed on 16 October 2022) (Appendix A). To evaluate sequence similarity, MEGAX pairwise distance analysis was implemented (https://www.megasoftware.net/, accessed on 27 September 2022). The generation of sequence logos was carried out using the WebLogo 3 server (http://weblogo.threeplusone.com, accessed on 27 September 2022) [64].

### 4.4. Determination of Antibacterial Activity of LF3872

The antibacterial activity of LF3872 against the studied test strains (*S. aureus, E. coli*, *Salmonella*, and *Campylobacter*) was determined by co-cultivation in TGVC medium at a temperature of 37 ± 1 °C for 48 h [112] with modifications. Briefly, for co-cultivation, 1 mL of LF3872 inoculum grown on MRC medium (10^9^ CFU/mL) and the test strain under study (10^7^–10^8^ CFU/mL) were introduced into 20 mL of TGVC medium. In the preliminary experiments, it was found that all strains (LF3872, *S. aureus*, *E. coli*, *Salmonella*, and *Campylobacter*) grew on TGVC medium. The counting of the cells of the test strains grown on TGVC medium in monoculture (control) and grown in the presence of lactobacilli on TGVC medium was carried out after 24 and 48 h. Aliquots of 1 mL were taken aseptically at 24 and 48 h; serially diluted; and spread onto TGVC agar for *S. aureus,* LB agar for *E. coli* and *Salmonella,* and BHI agar for *Campylobacter* cultivability. All the plates were incubated for 24–48 h at 37 °C under aerobic conditions for *S. aureus, E. coli,* and *Salmonella* and under microaerophilic conditions for *Campylobacter*; at the end of incubation, the colonies were counted and expressed as colony-forming units per milliliter (CFU/mL).

### 4.5. Co-aggregation Assay for Determination of Interactions between LF3872 and Proteobacteria Pathogens

Co-aggregation between LF3872 and strains of proteobacteria pathogens was determined by measuring the optical density (OD) at 600 nm according to [113]. The percentage of co-aggregation was calculated by the following equation:(1)$$\mathrm{Co-aggregation}\;\left(\%\right)=\frac{\left(\left(\mathrm{ODtarget}\;+\;\mathrm{ODLF}3872\right)-2\;\cdot \;\mathrm{ODmix}\right)}{\;\left(\mathrm{ODtarget}\;+\;\mathrm{ODLF}3872\right)}$$
where OD_target_, OD_LF3872_, and OD_mix_ represent the OD measure at 600 nm of the individual proteobacteria pathogen, LF3872, and their mixture after incubation for 2 h.

According to Ekmekci et al. [114], co-aggregation can be affected by pH. Hence, MES buffer (Sigma-Aldrich, Saint Louis, MO, USA) with a pH of 5 was used instead of PBS for cell suspension.

### 4.6. Preparation of CSLF3872

Cell-free culture supernatant was prepared as previously described [115], with modifications. Briefly, LF3872 was grown overnight in MRS broth under anaerobic conditions at 37 °C. The overnight culture was diluted to the concentration of 1 × 10^8^ CFU/mL in MRS broth and further grown anaerobically for 48 h. Cell-free culture supernatant was collected by centrifugation at 5000× *g* for 20 min at 4 °C; filter-sterilized using a 0.22 µm pore size filter (Millipore, Billerica, MA, USA); and concentrated by speed-vacuum drying (Rotational Vacuum Concentrator RVC2-18, Martin Christ, Osterode, Germany). The lyophilized sediment of CSLF3872 alone or together with the prebiotic was used to study the effectiveness of inhibiting the adhesion of proteobacteria pathogens (*E. coli*, *Salmonella*, and *Campylobacter*).

### 4.7. Assessment of Cytoplasmic Membrane Permeability by Measurement of Extracellular ATP in Indicator Bacteria

The indicator bacteria (*S. aureus*, *E. coli*, *Salmonella*, and *Campylobacter*) in the logarithmic phase were centrifuged and resuspended in PBS (pH 7.0) with OD600 = 1.0 (4 – 5 × 10^8^ CFU/mL). Bacteria were treated for 2.5 h at 37 °C in the presence of CSLF3872 (100µg/mL), and extracellular ATP levels after CSLF3872 treatment were detected by an ATP detection kit (Beyotime, Shanghai, China). Luminescence detection was performed using an Infinite 200 PRO microplate reader (Tecan, Männedorf, Switzerland).

### 4.8. In Vitro Caco-2 Bio Model of Immature Small Intestine to Study the Anti-Adhesive Properties of CSLF3872, Actigen Prebiotic, or Their Mixture against Proteobacteria Pathogens

Immortalized epithelial Caco-2 cells were used as an in vitro intestinal epithelial model. Caco-2 cell cultures are the most widely used cell lines for studying the properties of various drugs, probiotics, and prebiotics and predicting their impact on human and animal health, as well as for assessing their activity against pathogens [116,117,118,119,120]. In our experiments, Caco-2 cells were grown in monolayers of immature cells in Dulbecco’s Modified Eagle Medium (DMEM, Invitrogen, Carlsbad, CA, USA), with a layer of 80–100% cells supplemented with 20% heat-inactivated fetal bovine serum (FBS), 2 mM L-glutamine, penicillin (100 U/mL), and streptomycin (100 mg/mL) and kept in a CO_2_ incubator with 5% CO_2_. Immature Caco-2 cell monolayers were cultured for two days. The medium was replaced daily. Immature Caco-2 cell monolayers were used to study the anti-adhesive properties of CSLF3872; Actigen prebiotic (Alltech Inc., USA); or their mixture against proteobacteria pathogens *E. coli*, *Salmonella*, and *Campylobacter*. Suspensions of each strain with 5 × 10^7^ cells/mL were incubated in the presence of CSLF3872 at a concentration of 40 µg/mL, Actigen prebiotic at a concentration of 40 µg/mL, or their mixture (with CSLF3872 at concentration of 20 µg/mL and Actigen at concentration of 20 µg/mL) in PBS for 30 min. The strains of proteobacteria pathogens treated with CSLF3872, Actigen, or their mixture and the control group treated with PBS were then applied to the monolayers of Caco-2 cells. The plates were incubated for 1 h at 37 °C under 5% CO_2_. Caco-2 cell monolayers were washed three times with sterile PBS to remove unbound bacteria and CSLF3872, Actigen, or their mixture; fixed with methanol; stained with azure-eosin (Pan Eco, Russia); and examined under a Leica DM 4500B microscope (Leica, Calgary, AB, Canada). Adherent bacteria were quantified using the Leica IM modular applications system (Leica, Calgary, AB, Canada). The adhesion of bacterial cells to epithelial cells was expressed as the average number of adhered bacteria per epithelial Caco-2 cell.

### 4.9. ELISA

The concentration of human cytokines IL-8, IL-6, IL-10, and IL-12 in supernatants from cultured Caco-2 cells and IDCs was measured using ELISA kits from Biosource International, Carlsbad, CA, USA, whereas TNF-α and IL-1β were measured with ELISA kits from the State Research Institute of Highly Pure Biopreparations, St-Petersburg, Russia. The concentration of mouse TGF-β in supernatants from PP cells was measured with an ELISA kit from Biosource International; IFN-γ was measured with an ELISA kit from PharMingen, San Diego, CA, USA; and IgA from PP cells, the small intestine, and the colon was measured with an ELISA kit from Bethyl Laboratories, Inc., Montgomery, TX, USA, following the protocols provided by the manufacturers.

### 4.10. In Vitro Caco-2 Bio Model of Mature Small Intestine to Study the Effect of LF3872 on Innate Immunity

Caco-2 cells were grown in Dulbecco’s Modified Eagle Medium (DMEM, Invitrogen, Carlsbad, CA, USA) supplemented with 20% heat-inactivated fetal bovine serum (FBS), 2 mM L-glutamine, penicillin (100 U/mL), and streptomycin (100 mg/mL) and kept in a CO_2_ incubator with 5% CO_2_. The medium was replaced daily. Caco-2 cells reached confluence by day 15 (bio model of mature small intestine), at which point the morphological and functional differentiation was complete [121].

### 4.11. Acquisition of Human IDCs to Study the Effect of LF3872 on Innate Immunity

IDCs were obtained by the method proposed in [122], with modifications. Peripheral blood monocytes were isolated from the fresh heparinized blood of healthy donors in Ficoll-Hypaque (Sigma, St. Louis, MO, USA) by centrifugation with a density gradient; washed three times; and resuspended in RPMI 1640 (Sigma, St. Louis, MO, USA). Monocytes were additionally purified using magnetic beads (MACS, Miltenyi Biotec, Auburn, CA, USA). The magnetic beads contained antibodies immobilized to: CD3 (T-lymphocytes), CD7 (T/NK cells), CD16 (NK cells), CD19 (B-lymphocytes), CD56 (NK cells), CD123 (dendritic cells and basophils), and glycophorin A antibodies (erythrocytes and reticulocytes). CD14 expression on the MPC surface was determined by flow cytofluorometry. Monocytes were incubated at 4 °C 1 h with isotypic mouse IgG or CD14 monoclonal antibodies, and then with FITC-labeled anti-mouse IgG. The stained cells were washed twice; fixed in 1% paraformaldehyde; and analyzed on a FACSCalibur flow cytometer (Becton Dickinson, Franklin Lakes, NJ, USA). The obtained data were processed using WinMD12.8 software. According to the flow cytometry, the CD14 expression on the surface of the highly purified monocytes was 92.1 ± 6.4% (Appendix A). To obtain IDCs, monocytes in quantities of 1.5 × 10^6^ cells/mL were cultured on plastic mattresses (75 cm^2^) in RPMI 1640 (20 mL) containing IL 4 (500 IU/mL), GM-CSF (800 IU/mL), and 10% human blood serum for 6 days. On the sixth day, the obtained immature dandy cells were washed with fresh culture medium, and the CD14 expression on their surface was analyzed using cytofluorometry. IDCs, unlike primary peripheral blood monocytes, do not express CD14. According to the cytofluorometry, the IDCs used in the study did not express CD14 on their surface (Appendix A). IDCs were transplanted into 24-well plates. The sowing dose per well was 10^5^ cells/mL. The first experimental group of wells contained intact IDCs (first control group); the second group contained IDCs + LPS 10 ng/mL; the third group contained IDCs + 10^6^ CFU/mL LF3872; the fourth group contained intact IDCs (second control group); the fifth group contained IDCs + LPS 10 ng/mL; and the sixth group contained IDCs + LPS + 10^6^ CFU/mL LF3872. The plates were placed in a CO_2_ incubator for 24 h at 37 °C in an atmosphere containing 5% CO_2_. After 24 h of cultivation, the concentration of IL-10 and IL-12 in the supernatants was determined by the sandwich ELISA method.

### 4.12. LF3872 Effect Evaluation on Adaptive Immunity

The effect of LF3872 on adaptive immunity was assessed according to the production of TGF-β, IFN-γ, and IgA antibodies by immunocompetent cells of mouse PPs and the intestinal secretion of IgA antibodies. Two groups of BALB/c mice aged 6–8 weeks and weighing 25–30 g were included (bred in the experimental biological laboratory, EDITO Research Institute, N.N. Blokhin National Research Institute of Oncology, the Ministry of Health of Russia). Water and animal feed were provided ad libitum. The studies were conducted in accordance with the regulations on working with animals. The experimental group received a strain in drinking water (1.5 × 10^9^ cells/mL *L. fermentum*) for 5 consecutive days. The control group received clean drinking water for 5 consecutive days. After 5 days, the animals in the control and experimental groups were euthanized with CO_2_, PPs were extracted, and immunocompetent cells were isolated. The small intestine and colon were also extracted. The production of TGF-β, IFN-γ, and IgA by immunocompetent cells in the PPs and IgA production in the small intestine and colon were studied using the sandwich ELISA method.

### 4.13. Isolation of Immunocompetent Cells from Peyer’s Patches and Extraction of Small Intestine and Colon

PPs were isolated from the small intestine of BALB/c mice. The small intestine was excised, washed with a Krebs–Ringer solution, and cut along the mesentery line; then, PPs (oval nodular clusters of lymphoid tissue) were cut out of the ileum and placed in an RPMI-1640 medium containing 10% fetal calf serum, 0.2 M HEPES, and 25 U/mL collagenase. The cells were incubated at 37 °C on a magnetic stirrer for 10 min. The cell suspension was filtered through a nylon strainer and centrifuged at 500× *g* for 10 min, the supernatant was removed, RPMI-1640 medium was added to the precipitate, and centrifugation was repeated to wash the cells. The cell precipitate was resuspended in RPMI-1640 medium containing 10% fetal calf serum. The resulting cells of the PP lymphoid nodules at 5 × 10^5^ cells/mL were cultured in 96-well plates (Costar-Corning, New York, NY, USA). The small intestine and colon were preserved at −80 °C before thawing and removing the adipose tissue. Cooled sterile PBS solution was added according to a weight ratio of 1:9. The small intestine and colon were broken down and then centrifuged at 4 °C for 2000× *g* for 10 min. The supernatants were collected for assay measurement. Protein was measured in intestinal tissue using the Bradford method.

### 4.14. Statistical Analysis

The results were analyzed using one-way analysis of variance (ANOVA) and represented as the means ± standard errors of the means (SEM) of six independent experiments, tested in triplicate. Significance was evaluated by *t*-tests. Results were considered significant at *p* < 0.05.

## 5. Conclusions

LF3872 displayed a wide spectrum of antibacterial activity against antibiotic-resistant strains of Gram-positive and Gram-negative pathogens (*S. aureus*, *E. coli*, *Salmonella*, and *Campylobacter*). CSLF3872 induced cell damage in pathogens and ATP leakage. The mixture of CSLF3872 and Actigen prebiotic (Alltech Inc., Nicholasville, KY, USA) showed synergism in inhibiting the adhesion of proteobacteria pathogens to Caco-2 cells. LF3872 did not stimulate the production of pro-inflammatory cytokines IL-8, TNF-α, and IL-1β and slightly increased the IL-6 production in Caco-2 enterocytes. LF3872 stimulated the production of IL-10 and inhibited the production of IL-12 in human IDCs. The oral administration of LF3872 to mice stimulated the production of TGF-β, IFN-γ cytokines, and IgA antibodies by immunocompetent PPs cells and IgA antibodies in the intestine. LF3872 produced bacteriocin BLF3872, which belongs to the group of bacteriolysins that may destroy the peptidoglycan in the cell walls of bacterial pathogens. These results indicate the possibility of creating a synbiotic based on LF3872 and a prebiotic derived from *S. cerevisiae* cell wall components. Such innovative drugs and biologically active additives are necessary for the implementation of a strategy to reduce the spread of antibiotic-resistant strains of socially significant animal and human infections.

## Figures and Tables

**Figure 1 antibiotics-11-01437-f001:**
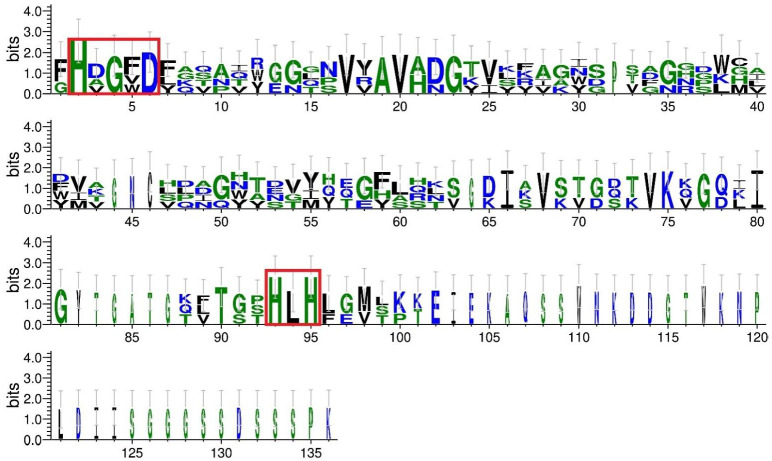
Logo motif representation for 280–404 region in BLF3872 bacteriocin corresponding to the peptidase M23 domain and peptidase M23 domain from enterolysin A, zoocin A, and zoocin A peptidase family M23. Logo profiles were generated using WebLogo 3 server [64]. Conserved motifs corresponding to the potential zinc binding site are highlighted.

**Figure 2 antibiotics-11-01437-f002:**
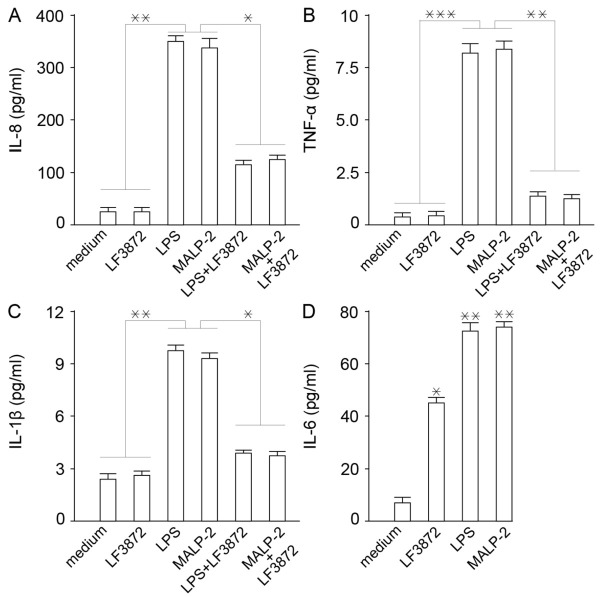
Regulation of (**A**) IL-8, (**B**) TNF-α, (**C**) IL1-1β, (**D**) IL-6 production induced by MALP-2 (TLR-2/6 receptor agonist) and LPS (TLR4 receptor agonist) in immature Caco-2 cells by LF3872. IL-8 level determined in supernatants cultured for 24 h, TNFα, IL-1β and IL-6 levels determined in supernatants cultured for 8 h in the presence of LF3872 strain. MALP-2 and LPS are as a control. Bars are means ± SEM. All data were representative of six independent experiments, tested in triplicate. (**A**) ** *p* < 0.01 medium, LF3872 vs. MALP-2 or LPS; * *p* < 0.05 MALP-2 vs. MALP-2+LF3872 or LPS vs. LPS+LF3872. (**B**) *** *p* < 0.001 medium, LF3872 vs. MALP-2 or LPS; ** *p* < 0.01 MALP-2 vs. MALP-2+LF3872 or LPS vs. LPS+LF3872. (**C**) ** *p* < 0.01 medium, LF3872 vs. MALP-2 or LPS; * *p* < 0.05 MALP-2 vs. MALP-2+LF3872 or LPS vs. LPS+LF3872. (**D**) ** *p* < 0.01 medium vs. MALP-2 or LPS; * *p* < 0.05 medium vs. LF3872.

**Figure 3 antibiotics-11-01437-f003:**
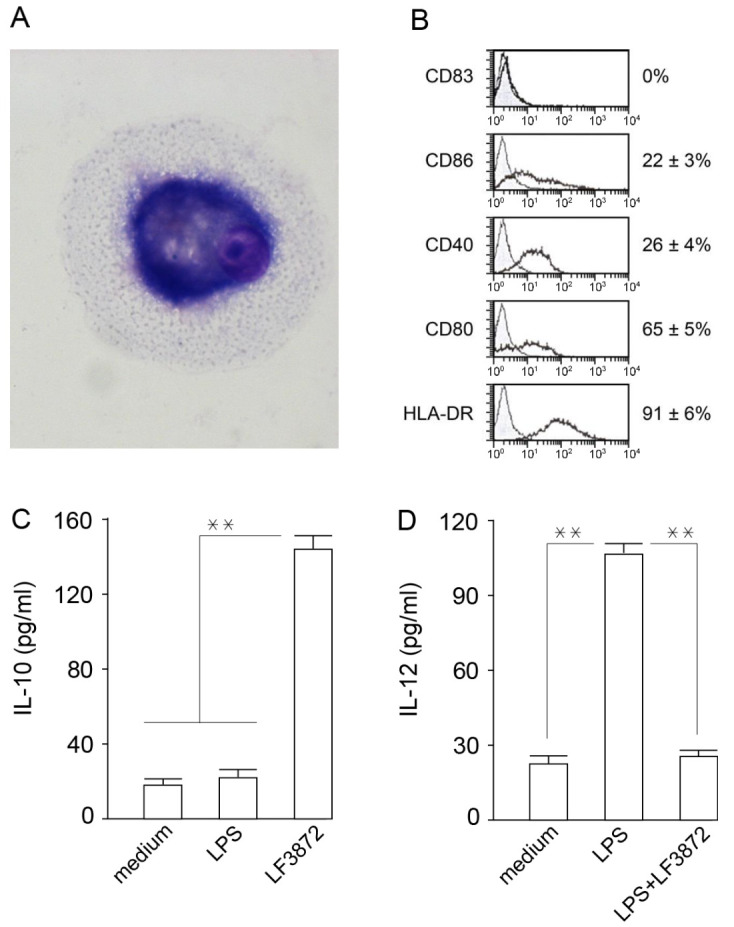
LF3872 regulates the production of IL-10 and IL-12 in human IDCs. (**A**) IDC—light microscopy after azure-eosin staining; (**B**) phenotype of IDCs; (**C**) LF3872 stimulates the production of IL-10; (**D**) LF3872 inhibits the production of IL-12. Notes: the amount of IDCs was 10^5^ cells/mL; the ratio of IDCs/lactobacilli was 1:10; and the cultivation time was 24 h. ** *p* < 0.01. Figure 3C—IL-10 production in IDCs + LF3872 vs. medium, or LPS; Figure 3D—IL-12 production in IDCs + LPS vs. medium or LPS + LF3872.

**Figure 4 antibiotics-11-01437-f004:**
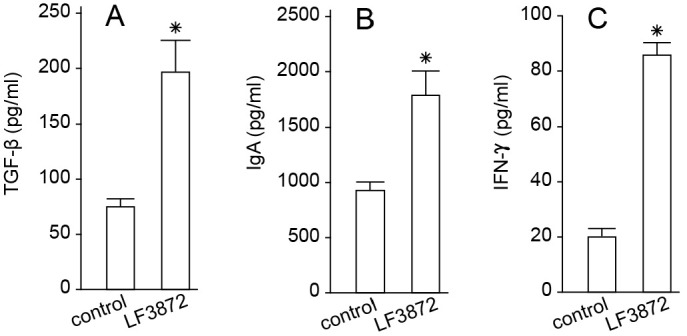
LF3872 stimulates the production of (**A**) TGF-β, (**B**) IgA, and (**C**) IFN-γ in Peyer’s patches of mice. * *p* < 0.05. Data are presented as the means ± SEM of three independent experiments, tested in triplicate.

**Figure 5 antibiotics-11-01437-f005:**
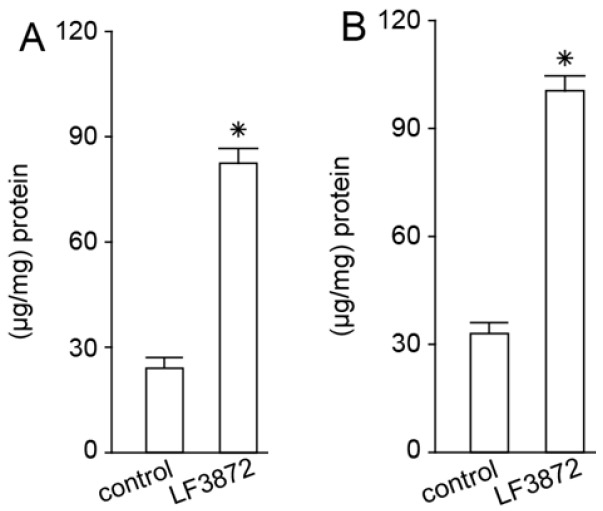
LF3872 stimulates the production of IgA in the small intestine (**A**) and colon (**B**) of mice. * *p* < 0.05. Data are presented as the means ± SEM of three independent experiments, tested in triplicate.

**Table 1 antibiotics-11-01437-t001:** Comparison of class III bacteriocin profile of peptidase M23 and BLF3872 bacteriocin.

Protein Name	UniProt ID	Microorganism	Peptidase M23 Domain Position, a.a. Number	Domain Length	% Identity	Pairwise Distances
BLF3872		LF3872	280–440	161		
enterolysin A	Q9F8B0	*E. faecalis*	62–154	93	35	1.00
zoocin A	O54308	*S. zooepidemicus*	44–138	95	33	1.22
zoocin A peptidase family M23	A5FNQ4	*F. johnsoniae*	50–105	56	33	1.12

**Table 2 antibiotics-11-01437-t002:** Antagonistic activity of LF3872 against *S. aureus* strains.

Strain	0 h			24 h			48 h		
	C ^1^	JC ^2^	HJC ^3^	C ^1^	JC ^2^	HJC ^3^	C ^1^	JC ^2^	HJC ^3^
*S. aureus* 8325-4	2 *×* 10^8^	1 × 10^8^	1 *×* 10^8^	2 *×* 10^8^	<10^2^	3 *×* 10^8^	4 *×* 10^8^	<10^2^	4 *×* 10^8^
*S. aureus* IIE CI-SA 1246	5 *×* 10^8^	4 × 10^8^	4 *×* 10^8^	5 *×* 10^8^	<10^2^	5 *×* 10^8^	6 *×* 10^8^	<10^2^	6 *×* 10^8^

^1^ Control, number of *S. aureus* cells in monoculture (CFU/mL). ^2^ Number of *S. aureus* cells in co-cultivation with LF3872 (CFU/mL). ^3^ Number of *S. aureus* cells in co-cultivation with heat-treated (56 °C, 30 min) LF3872 (CFU/mL). Data are representative of six independent experiments, tested in triplicate.

**Table 3 antibiotics-11-01437-t003:** Antagonistic activity of LF3872 against *E. coli* strains.

Strain	0 h			24 h			48 h		
	C ^1^	JC ^2^	HJC ^3^	C ^1^	JC ^2^	HJC ^3^	C ^1^	JC ^2^	HJC ^3^
*E. coli* ATCC BAA198	2 *×* 10^7^	1 *×* 10^7^	1 *×* 10^7^	4 *×* 10^7^	2 *×* 10^4^	4 *×* 10^7^	6 *×* 10^7^	5 *×* 10^3^	6 *×* 10^7^
*E. coli* ATCC BAA 204	7 *×* 10^6^	8 *×* 10^6^	7 *×* 10^6^	9 *×* 10^6^	1 *×* 10^4^	9 *×* 10^6^	2 *×* 10^7^	2 *×* 10^3^	3 *×* 10^7^
*E. coli* ATCC BAA 2326	3 *×* 10^7^	3 *×* 10^7^	2 *×* 10^7^	4 *×* 10^7^	3 *×* 10^4^	4 *×* 10^7^	5 *×* 10^7^	6 *×* 10^3^	5 *×* 10^7^
*E. coli* IIE Br 5164	5 *×* 10^7^	4 *×* 10^7^	5 *×* 10^7^	6 *×* 10^7^	5 *×* 10^4^	6 *×* 10^7^	8 *×* 10^7^	7 *×* 10^3^	8 *×* 10^7^
*E. coli* IIE Br 5372	8 *×* 10^6^	8 *×* 10^6^	7 *×* 10^6^	9 *×* 10^6^	5 *×* 10^4^	9 *×* 10^6^	1 *×* 10^7^	2 *×* 10^3^	2 *×* 10^7^
*E. coli* IIE Pi 5548	1 *×* 10^7^	2 *×* 10^7^	1 *×* 10^7^	3 *×* 10^7^	2 *×* 10^4^	3 *×* 10^7^	4 *×* 10^7^	3 *×* 10^3^	4 *×* 10^7^
*E. coli* IIE Co 5622	4 *×* 10^7^	4 *×* 10^7^	3 *×* 10^7^	5 *×* 10^7^	6 *×* 10^4^	6 *×* 10^7^	7 *×* 10^7^	6 *×* 10^3^	7 *×* 10^7^
*E. coli* IIE Hu 4326	3 *×* 10^7^	3 *×* 10^7^	4 *×* 10^7^	4 *×* 10^7^	4 *×* 10^4^	5 *×* 10^7^	8 *×* 10^7^	9 *×* 10^3^	8 *×* 10^7^
*E. coli* ATCC E 2348/69	7 *×* 10^6^	7 *×* 10^6^	7 *×* 10^6^	8 *×* 10^6^	2 *×* 10^4^	8 *×* 10^6^	1 *×* 10^7^	2 *×* 10^3^	2 *×* 10^7^
*E. coli* ATCC E 31705	4 *×* 10^7^	4 *×* 10^7^	5 *×* 10^7^	6 *×* 10^7^	3 *×* 10^4^	6 *×* 10^7^	7 *×* 10^7^	4 *×* 10^3^	7 *×* 10^7^

^1^ Control, number of *E. coli* cells in monoculture (CFU/mL). ^2^ Number of *E. coli* cells in co-cultivation with LF3872 (CFU/mL). ^3^ Number of *E. coli* cells in co-cultivation with heat-treated (56 °C, 30 min) LF3872 (CFU/mL). Data are representative of six independent experiments, tested in triplicate.

**Table 4 antibiotics-11-01437-t004:** Antagonistic activity of LF3872 against *Salmonella* strains.

Strain	0 h			24 h			48 h		
	C ^1^	JC ^2^	HJC ^3^	C ^1^	JC ^2^	HJC ^3^	C ^1^	JC ^2^	HJC ^3^
*S.* Enteritidis ATCC 13076	5 × 10^7^	5 × 10^7^	4 × 10^7^	6 × 10^7^	2 × 10^4^	6 × 10^7^	7 × 10^7^	6 × 10^3^	7 × 10^7^
*S.* Enteritidis ATCC 4931	2 × 10^7^	1 × 10^7^	2 × 10^7^	3 × 10^7^	5 × 10^4^	3 × 10^7^	4 × 10^7^	3 × 10^3^	5 × 10^7^
*S.* Enteritidis IIE Egg 6215	3 × 10^7^	3 × 10^7^	2 × 10^7^	4 × 10^7^	7 × 10^4^	3 × 10^7^	4 × 10^7^	5 × 10^3^	4 × 10^7^
*S.* Typhimurium ATCC 700720	8 × 10^6^	9 × 10^6^	8 × 10^6^	1 × 10^7^	1 × 10^4^	9 × 10^6^	2 × 10^7^	6 × 10^3^	3 × 10^7^
*S.* Typhimurium ATCC 14028	4 × 10^7^	4 × 10^7^	3 × 10^7^	5 × 10^7^	3 × 10^4^	5 × 10^7^	6 × 10^7^	5 × 10^3^	7 × 10^7^
*S.* Typhimurium IIE Br 6458	6 × 10^7^	6 × 10^7^	5 × 10^7^	8 × 10^7^	6 × 10^4^	8 × 10^7^	9 × 10^7^	7 × 10^3^	9 × 10^7^

^1^ Control, number of *Salmonella* cells in monoculture (CFU/mL). ^2^ Number of *Salmonella* cells in co-cultivation with LF3872 (CFU/mL). ^3^ Number of *Salmonella* cells in co-cultivation with heat-treated (56 °C, 30 min) LF3872 (CFU/mL). Data are representative of six independent experiments, tested in triplicate.

**Table 5 antibiotics-11-01437-t005:** Antagonistic activity of LF3872 against *C. jejuni* strains.

Strain	0 h			24 h			48 h		
	C ^1^	JC ^2^	HJC ^3^	C ^1^	JC ^2^	HJC ^3^	C ^1^	JC ^2^	HJC ^3^
*C. jejuni* ATCC 29428	4 × 10^7^	4 × 10^7^	3 × 10^7^	5 × 10^7^	3 × 10^5^	5 × 10^7^	6 × 10^7^	8 × 10^4^	6 × 10^7^
*C. jejuni* IIE Br 7154	3 × 10^7^	3 × 10^7^	2 × 10^7^	4 × 10^7^	2 × 10^5^	4 × 10^7^	5 × 10^7^	7 × 10^4^	5 × 10^7^
*C. jejuni* IIE Br 7365	6 × 10^7^	6 × 10^7^	6 × 10^7^	7 × 10^7^	4 × 10^5^	6 × 10^7^	8 × 10^7^	9 × 10^4^	8 × 10^7^
*C. jejuni* IIE Br 7548	2 × 10^7^	2 × 10^7^	3 × 10^7^	3 × 10^7^	2 × 10^5^	4 × 10^7^	4 × 10^7^	6 × 10^4^	5 × 10^7^

^1^ Control, number of *C. jejuni* cells in monoculture (CFU/mL). ^2^ Number of *C. jejuni* cells in co-cultivation with LF3872 (CFU/mL). ^3^ Number of *C. jejuni* cells in co-cultivation with heat-treated (56 °C, 30 min) LF3872 (CFU/mL). Data are representative of six independent experiments, tested in triplicate.

**Table 6 antibiotics-11-01437-t006:** Extracellular ATP levels in tested bacteria treated with CSLF3872.

Strain	Boiled CSLF3872 ^1^	CSLF3872 ^2^
*S.aureus* IIE 8325-4	6.0 ± 2.3	358.7 ± 14.3 ***
*S.aureus* IIE CI-SA 1246	5.4 ± 1.5	360.5 ± 12.6 ***
*E. coli* ATCC BAA198	4.1 ± 0.8	37.3 ± 5.0 **
*E. coli* ATCC BAA 204	3.9 ± 0.5	42.6 ± 3.7 **
*E. coli* ATCC BAA 2326	4.3 ± 0.7	45.4 ± 3.4 **
*E. coli* IIE Br 5164	3.0 ± 0.5	39.2 ± 3.7 **
*E. coli* IIE Br 5372	5.4 ± 1.2	46.7 ± 2.5 **
*E. coli* IIE Pi 5548	4.6 ± 0.8	38.4 ± 4.2 **
*E. coli* IIE Co 5622	3.5 ± 0.7	34.8 ± 3.5 **
*E. coli* IIE Hu 4326	4.3 ± 0.9	46.5 ± 4.0 **
*E. coli* ATCC E 2348/69	3.7 ± 0.8	42.7 ± 3.4 **
*E. coli* ATCC E 31705	4.5 ± 1.1	39.5 ± 2.7 **
*S.* Enteritidis ATCC 13076	3.8 ± 0.9	45.0 ± 3.6 **
*S.* Enteritidis ATCC 4931	4.7 ± 1.0	48.4 ± 3.2 **
*S.* Enteritidis IIE Egg 6215	4.6 ± 1.2	46.2 ± 3.8 **
*S.* Typhimurium ATCC 700720	4.0 ± 0.9	43.5 ± 5.1 **
*S.* Typhimurium ATCC 14028	5.3 ± 1.4	46.4 ± 2.7 **
*S.* Typhimurium IIE Br 6458	5.0 ± 1.2	47.8 ± 3.0 **
*C. jejuni* ATCC 29428	5.7 ± 0.6	21.5 ± 1.2 *
*C. jejuni* IIE Br 7154	5.9 ± 0.8	18.6 ± 1.5 *
*C. jejuni* IIE Br 7365	5.6 ± 0.7	22.7 ± 2.0 *
*C. jejuni* IIE Br 7548	6.0 ± 0.8	18.9 ± 1.8 *

^1, 2^ Concentration of ATP (nm/OD). * *p* < 0.05 extracellular ATP levels in *C. jejuni* strains (control, boiled CSLF3872) vs. native CSLF3872; ** *p* < 0.01 extracellular ATP levels in *E. coli* and *Salmonella* strains (control, boiled CSLF3872) vs. native CSLF3872; *** *p* < 0.001 extracellular ATP levels in *S. aureus* strains (control, boiled CSLF3872) vs. native CSLF3872. All data are representative of six independent experiments, tested in triplicate.

**Table 7 antibiotics-11-01437-t007:** Co-aggregative abilities of LF3872 strain with proteobacteria pathogens.

Strain	20 °C, pH 7.0 ^1^	20 °C, pH 5.0 ^2^
*E. coli* ATCC BAA198	40.2 ± 5.3	74.5 ± 6.3 *
*E. coli* ATCC BAA 204	37.6 ± 4.1	65.9 ± 7.1 *
*E. coli* ATCC BAA 2326	34.5 ± 4.2	75.3 ± 7.2 *
*E. coli* IIE Br 5164	43.3 ± 4.5	67.8 ± 6.5 *
*E. coli* IIE Br 5372	38.4 ± 4.1	69.4 ± 5.1 *
*E. coli* IIE Pi 5548	42.6 ± 5.5	72.5 ± 5.6 *
*E. coli* IIE Co 5622	37.8 ± 4.6	64.3 ± 5.4 *
*E. coli* IIE Hu 4326	44.3 ± 5.4	69.2 ± 5.0 *
*E. coli* ATCC E 2348/69	41.5 ± 3.7	64.5 ± 5.8 *
*E. coli* ATCC E 31705	42.6 ± 4.2	64.2 ± 5.6 *
*S.* Enteritidis ATCC 13076	38.4 ± 5.1	71.6 ± 6.3 *
*S.* Enteritidis ATCC 4931	39.6 ± 4.8	68.8 ± 6.5 *
*S.* Enteritidis IIE Egg 6215	42.1 ± 6.5	68.6 ± 5.4 *
*S.* Typhimurium ATCC 700720	38.5 ± 6.2	75.8 ± 6.7 *
*S.* Typhimurium ATCC 14028	40.7 ± 4.5	72.4 ± 6.5 *
*S.* Typhimurium IIE Br 6458	39.4 ± 4.2	74.6 ± 5.3 *
*C. jejuni* ATCC 29428	32.5 ± 3.4	65.8 ± 7.1 *
*C. jejuni* IIE Br 7154	38.3± 4.2	67.2 ± 6.4 *
*C. jejuni* IIE Br 7365	35.6± 3.7	70.5 ± 5.9 *
*C. jejuni* IIE Br 7548	36.4± 3.5	66.4 ± 5.7 *

^1, 2^ Co-aggregation with LF3872 (%). The percentage of co-aggregation was measured at pH 7.0 in PBS buffer and pH 5.0 in MES buffer. * *p* < 0.05 co-aggregation of LF3872 strain with proteobacteria pathogens at pH 5.0 vs. pH 7.0. All data are representative of six independent experiments, tested in triplicate.

**Table 8 antibiotics-11-01437-t008:** Synergism of CSLF3872 and Actigen prebiotic in inhibiting the adhesion of proteobacteria pathogens (*E. coli*, *Salmonella*, *Campylobacter*) to Caco-2 cells.

Strain	PBS, Control	CSLF3872 ^1^	Actigen ^2^	MIXT ^3^
*E. coli* ATCC BAA198	36.3 ± 1.8	7.5 ± 1.1 ^**^	5.6 ± 0.8 ^**^	0.84 ± 0.05 ^***^
*E. coli* ATCC BAA 204	34.8 ± 1.6	8.4 ± 1.4 ^**^	5.3 ± 0.5 ^**^	0.60 ± 0.07 ^***^
*E. coli* ATCC BAA 2326	35.5 ± 1.7	8.7 ± 0.9 ^**^	5.5 ± 0.4 ^**^	0.72 ± 0.06 ^***^
*E. coli* IIE Br 5164	36.4 ± 1.3	8.3 ± 1.0 ^**^	6.3 ± 0.7 ^**^	0.74 ± 0.05 ^***^
*E. coli* IIE Br 5372	31.7 ± 1.5	7.8 ± 1.1 ^**^	4.8 ± 0.5 ^**^	0.59 ± 0.04 ^***^
*E. coli* IIE Pi 5548	34.5 ± 1.2	9.3 ± 1.5 ^**^	4.7 ± 0.6 ^**^	0.68 ± 0.05 ^***^
*E. coli* IIE Co 5622	35.2 ± 1.7	8.4 ± 1.2 ^**^	4.9 ± 0.5 ^**^	0.60 ± 0.04 ^***^
*E. coli* IIE Hu 4326	33.6 ± 1.5	7.6 ± 1.3 ^**^	4.6 ± 0.4 ^**^	0.75 ± 0.03 ^***^
*E. coli* ATCC E 2348/69	32.5 ± 1.6	6.4 ± 1.0 ^**^	5.3 ± 0.6 ^**^	0.81 ± 0.07 ^***^
*S.* Enteritidis ATCC 13076	24.7 ± 1.3	8.3 ± 1.2 ^**^	4.5 ± 0.8 ^**^	0.69 ± 0.04 ^***^
*S.* Enteritidis ATCC 4931	29.5 ± 1.2	9.8 ± 1.0 ^**^	6.4 ± 0.7 ^**^	0.70 ± 0.05 ^***^
*S.* Enteritidis IIE Egg 6215	28.2 ± 1.5	7.4 ± 0.9 ^**^	4.9 ± 0.5 ^**^	0.85 ± 0.04 ^***^
*S.* Typhimurium ATCC 700720	25.5 ± 1.4	8.2 ± 1.6 ^**^	5.2 ± 0.7 ^**^	0.83 ± 0.05 ^***^
*S.* Typhimurium ATCC 14028	27.4 ± 1.6	6.3 ± 1.1 ^**^	4.9 ± 0.5 ^**^	0.72 ± 0.06 ^***^
*S.* Typhimurium IIE Br 6458	28.4 ± 1.3	8.4 ± 1.5 ^**^	5.4 ± 0.6 ^**^	0.75 ± 0.03 ^***^
*C. jejuni* ATCC 29428	30.6 ± 1.5	7.9 ± 1.2 ^**^	17.4 ± 1.2 ^*^	0.80 ± 0.04 ^***^
*C. jejuni* IIE Br 7154	31.4 ± 1.7	4.2 ± 0.9 ^**^	16.5 ± 1.1 ^*^	0.65 ± 0.03 ^***^
*C. jejuni* IIE Br 7365	28.4 ± 1.9	7.4 ± 1.3 ^**^	16.9 ± 1.4 ^*^	0.49 ± 0.03 ^***^
*C. jejuni* IIE Br 7548	30.5 ± 1.6	5.7 ± 0.8 ^**^	17.5 ± 1.2 ^*^	0.72 ± 0.05 ^***^

^1^ Lyophilized CSLF3872 (concentration: 40 µg/mL). ^2^ Actigen prebiotic (Alltech Inc., USA) from *S. cereisiae* (concentration: 40 µg/mL). ^3^ Mixture of CSLF3872 (concentration: 20 µg/mL) and Actigen prebiotic (concentration: 20 µg/mL). * *p* < 0.05 adhesion of *Campylobacter* strains to Caco-2 alone vs. adhesion of *Campylobacter* strains to Caco-2 + Actigen; ** *p* < 0.01 adhesion of *E. coli*, *Salmonella*, and *Campylobacter* strains to Caco-2 alone vs. adhesion of *E. coli*, *Salmonella*, and *Campylobacter* strains to Caco-2 + CSLF3872 and adhesion of *E. coli* and *Salmonella* strains to Caco-2 alone vs. adhesion of *E. coli*, *Salmonella* strains to Caco-2 + Actigen; *** *p* < 0.001 adhesion of *E. coli*, *Salmonella*, and *Campylobacter* strains to Caco-2 alone vs. adhesion of *E. coli*, *Salmonella*, and *Campylobacter* strains to Caco-2 + CSLF3872 + Actigen. Data are presented as the means ± SEM of six independent experiments, tested in triplicate.

**Table 9 antibiotics-11-01437-t009:** Microorganisms used in this study.

Microorganism	Strain	Growth Conditions
*L. fermentum*	IIE ^1^ 3872	MRS ^a^ 37 °C in CO_2_ incubator, 10% CO_2_ or anaerobically 48 h
*S. aureus*	8325-4 ^2^ Antibiotic resistance GEN	TGVC ^b^ 37 °C aerobically 24 h
*S. aureus*	IIE CI-SA 1246 ^3^ Antibiotic resistance AMP	The same
*E. coli*	ATCC ^4^ BAA198 ESBL ^5^ Type TEM-26	LB ^c^ 37 °C aerobically 18 h
*E. coli*	ATCC BAA 204 ESBL Type SHV-2	The same
*E. coli*	ATCC BAA 2326 ESBL Type CTX-M-15	The same
*E. coli*	IIE Br ^6^ 5164 ESBL Type SHV	The same
*E. coli*	IIE Br 5372 ESBL Type CTX-M	The same
*E. coli*	IIE Pi ^7^ 5548 ESBL Type CTX-M	The same
*E. coli*	IIE Co ^8^ 5622 ESBL Type CTX-M	The same
*E. coli*	IIE Hu ^9^ 4326 ESBL Type CTX-M	The same
*E. coli*	ATCC E 2348/69 (EPEC) ^10^	The same
*E. coli*	ATCC E 31705 (ETEC) ^11^	The same
*S*. Enteritidis	ATCC 13076	BHI ^d^ 37 °C aerobically 18 h
*S*. Enteritidis	ATCC 4931	The same
*S*. Enteritidis	IIE Egg ^12^ 6215 Antibiotic resistance NAL/AMP	The same
*S.* Typhimurium	ATCC 700720	The same
*S.* Typhimurium	ATCC 14028	The same
*S.* Typhimurium	IIE Br 6458 Antibiotic resistance NAL/AMP/TET	The same
*C. jejuni*	ATCC 29428	BHI 37 °C microaerobically (5% O_2_, 10% CO_2_, 85% N_2_) 18 h
*C. jejuni*	IIE Br 7154 Antibiotic resistance CIP/NAL/STR	The same
*C. jejuni*	IIE Br 7365 Antibiotic resistance CIP/NAL/STR/TET	The same
*C. jejuni*	IIE Br 7548 Antibiotic resistance CIP/NAL/STR/TET/ERY	The same

^1^ Collection of Microorganisms at the Institute of Immunological Engineering (IIE), Department of Biochemistry of Immunity and Biodefence, Lyubuchany, Moscow Region, Russia. ^2^ Collection of Microorganisms at the National Research Center of Epidemiology and Microbiology named after Academician N.F.Gamaleya of the Ministry of Health of the Russian Federation. ^3^ Clinical isolate from the milk of a woman with mastitis. ^4^ American Type Culture Collection, Manassas, VA, USA. ^5^ Extended-spectrum beta-lactamase [109,110]. ^6^ Isolate from fresh broiler chicken meat. ^7^ Isolate from piglet intestine. ^8^ Isolate from the udder of a cow with mastitis. ^9^ Clinical isolate from the urine of a woman with cystitis. ^10^ Enteropathogenic *E. coli*. ^11^ Enterotoxigenic *E. coli*. ^12^ Isolate from chicken eggs. ^a^ Man–Rogosa–Sharp (MRS) broth or agar-containing plates (HiMedia, Hyderabad, India). ^b^ TGVC broth or agar (HiMedia, India). ^c^ Luria–Bertrani medium (LB) broth or agar (HiMedia, India). ^d^ Brain heart infusion (BHI) broth supplemented with 0.5% yeast extract or agar containing BHI plates. Antibiotic resistance: NAL—nalidixic acid; AMP—ampicillin; TET—tetracycline; CIP—ciprofloxacin; STR—streptomycin; ERY—erythromycin; GEN—gentamicin.

## Data Availability

Not applicable.

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
