# Peer review of "Limosilactobacillus fermentum Strain 3872: Antibacterial and Immunoregulatory Properties and Synergy with Prebiotics against Socially Significant Antibiotic-Resistant Infections of Animals and Humans"

_antibiotics, 2022, doi:10.3390/antibiotics11101437_

Round 1

Reviewer 1 Report

The manuscript is well-written and the obtained findings are properly described in the context of the published literature. I only have a few comments.

-The method and materials should come first then followed by the results and discussion.  

-4.2. Identification of BLF3872 bacteriocin produced by LF3872
The percentages of nucleotide similarity of the nucleotide sequences were not calculated.

-Line 476, 479 483, 486, Check the usage of future tense "will be".

Author Response

Reviewer #1:

  1. The method and materials should come first then followed by the results and discussion.

Response 1:

Sections of the article were arranged in order according to the instructions for authors of the journal Antibiotics.

  1. 4.2. Identification of BLF3872 bacteriocin produced by LF3872

The percentages of nucleotide similarity of the nucleotide sequences were not calculated.

Response 2:

The percentage of nucleotide similarity was added to the section 4.2.

  1. Line 476, 479 483, 486, Check the usage of future tense "will be".

Response 3:

Done

Reviewer 2 Report

The study done by Abramov et al demonstrated the antibacterial properties of LF3872 using test cultures of antibiotic-resistant Gram-positive and Gram-negative pathogens. G+ pathogens were found to be highly sensitive to the bacteriolytic action of LF3872, while G- pathogens were more resistant to the bacteriolytic action of LF3872. Cell-free culture supernatant of LF3872 induced cell damage of G+ and G- test cultures and ATP leakage. In in vitro experiments, it was found that LF3872 and the Actigen prebiotic exhibited synergistic anti-adhesive activity against G- pathogens. LF3872 has found to have immunoregulatory properties: it inhibited the lipopolysaccharide-induced production of proinflammatory cytokines IL-8, IL-1β, and TNF-α by a monolayer of Caco-2 cells, inhibits the production of IL-12 and stimulates the production of IL-10 by immature human dendritic cells, stimulates production of the TGF-β, IgA and IFN-γ by the immunocompetent cells of intestinal Peyer’s patches (PPs) of mice. The study is very interesting and have great scientific value; the manuscript is well written; the results are clearly presented; however, the discussion part needs improvement. The manuscript needs some minor corrections.

1.     Please maintain the order of results sections in accordance with the order of materials and methods sections (i.e., switch sections 2.7 and 2.6 in the result part).

2.     Replace Fig. 1 with one with higher resolution

3.     I recommend adding a microscopical image to the results (section 2.8) clarifying the adhesion of bacterial cells to Caco-2 cell.

4.     Please justify why the adhesion of G+ pathogens was not assessed as well?

5.     Please justify why the effect of LPS + LF3872 and MALP-2 + LF3872 on the expression level of IL-6 was not studied?

6.     Line 619. “in” is duplicated.

7.     Please include ethics committee approval code/number for animal study.

8.     The discussion part contains too many flaws; too much information, which I prefer to be transferred to introduction section, and it is repetition of results rather than comparing them to previous studies. The discussion part needs to be rewritten and add previous works.

Reviewer 3 Report

The Abramov et al. studied the antibacterial, co-aggregative, immunoregulatory properties of LF3872 and synergism of anti-adhesive activity of Actigen prebiotic with cell-free culture supernatant (CS) of LF3872. Additionally, they further investigated the immunoregulatory properties of LF3872 in Caco and DC cells and in mice. The work is well done and the conclusion is very useful for future further development. However, some conclusions are not rigorous and may need more data to support. I have a few comments for further improvement.

1. LF3872 was originally isolated from milk of a healthy woman. Add complete whole genome sequencing of the LF3872 strain revealed a gene encoding a unique bacteriocin. I assume that some studies about LF3872 have been done previously. The authors only cited one related reference in the introduction. Any more existing studies about LF3872? The authors should give more details and background of LF3872 in the introduction.

2. The antibacterial properties of LF3872 were studied using the test cultures of antibiotic-resistant Gram-positive (G+) and Gram-negative (G-) pathogens. G+ pathogens were highly sensitive to the bacteriolytic action of LF3872, while G- pathogens were more resistant to the bacteriolytic action of LF3872 compared to G+ pathogens. In these co-culture experiments (Tables 2-5), I assume that the culture medium is quite different between LF3872 and various G+ or G- pathogens. Do the authors have the data to demonstrate that LF3872 culture medium has no effect on the inhibition or lysis of various pathogens, like boiled L3872 culture as control?

3. Based on the data that LF3872 regulates cytokine production in intestinal Caco-2 cells, the authors claimed that “These data indicate that LF3872 interacts with TLR2/6 and TLR4 receptors of innate immunity expressed on the surface of Caco-2 enterocytes of the intestine”. This could happen but the conclusion is not rigorous. For example, it’s also possible that LF3872 binds to LPS or MALP-2. More data are needed to support this statement.

4. “LF3872 regulates cytokine production in Peyer’s patches of mice. The LF3872 regulates adaptive immunity. When administered orally to mice, the LF3872 stimulates the production of TGF-β, IgA antibodies and IFN-γ by immunocompetent Peyer’s patches (PPs) cells.” More details are needed in this section. What data support the regulation of adaptive immunity by LF3872? Why test immunocompetent Peyer’s patches (PPs) cells after oral administration? Any sera test? What does the production of TGF-β, IgA antibodies and IFN-γ by PPs mean here?

Reviewer 4 Report

In this manuscript, the antibacterial and immunoregulatory activity of Limosilactobacillus fermentum 3872 strain is evaluated.

The following comments are made:

1. Line 39. You can remove (G+) and (G-) from all the text, that abbreviation is not used.

2. Line 71. Put paragraph reference.

3. Line 75. Indicate which strain it is.

4. Line 86. “TEM, SHV, and CTX-M-15”. Put what the abbreviations mean.

5. Line 144. Explain what Actigen prebiotic is, characteristics, origin, etc.

6. Line 153. It is a very low percentage of identity with class III bacteriocins. What arguments do you have to consider BLF3872 of that class?

7. In the Results section you are mixing Discussion. You can only put the results and then discuss them.

8. Line 183-185. The characteristics of the origin of the strains can be put in Table 9.

9. Table 2, Table 3, Table 4, and Table 5. Put what C and JC mean.

10. Table 2. How you explain that there was no increase in CFU/mL in the control after 48h? The same happens in Tables 3, 4 and 5.

11. Line 252. In the Discussion section, discuss the importance between the difference of pH 7 and pH 5

12. Table 8. What is the explanation for putting *p<0.05; **p<0.05 and ***p<0.05, if they are the same?

13.Line 297-299. Because you ensure that LF3872 binds to receptors? There may be other explanations. It's just a possibility, discuss it.

14. Lines 313-324. Put in which figure these results are seen.

15. Line 424. Put what MOS means.

16. Line 483. Put what OM means.

17. Line 559. Put full CS.

18. Line 572. How many CFU/mL is that?

19. Line 603. Put what IDC means.

20. Line 618. Put full DC

21. Line 642. Put what LPS means

22. Line 651 Put what PPs mean.

23. Sections 4.12 and 4.13 have the same title, explain the difference or correct the title.

24. The conclusions have nothing to do with the results obtained. Draw conclusions about your work.

Round 2

Reviewer 4 Report

1. To keep an adequate use of the nomenclature in microbiology, eliminate G+ and G- and put Gram-positive and Gram-negative in all the text.

2. Put answer 5 regarding Actigen prebiotic in the text.

3. Lines 148-151. Put in the Introduction or in the Discussion, not in Results.

4. Put a final conclusion of the importance of your work.

Author Response

Reviewer #4 (Round 2):

  1. To keep an adequate use of the nomenclature in microbiology, eliminate G+ and G- and put Gram-positive and Gram-negative in all the text.

Response 1:

Replacement made throughout the text of the article.

  1. Put answer 5 regarding Actigen prebiotic in the text.

Response 2:

Answer added to the Discussion section.

  1. Lines 148-151. Put in the Introduction or in the Discussion, not in Results.

Response 3:

Information moved to the Introduction section.

  1. Put a final conclusion of the importance of your work.

Response 4:

A final conclusion about the importance of our work has been added to the Conclusion section.